# Fungal Pathogen Emergence: Investigations with an *Ustilago* *maydis* × *Sporisorium reilianum* Hybrid

**DOI:** 10.3390/jof7080672

**Published:** 2021-08-20

**Authors:** Emilee R. M. Storfie, Barry J. Saville

**Affiliations:** 1Department of Agricultural, Food, and Nutritional Science, University of Alberta, Edmonton, AB T6G 2R3, Canada; storfie@ualberta.ca; 2Environmental and Life Sciences Graduate Program, Trent University, Peterborough, ON K9J 7B8, Canada; 3Forensic Science Program, Trent University, Peterborough, ON K9J 7B8, Canada

**Keywords:** *Ustilago maydis*, *Sporisorium reilianum*, fungal hybridization, pathogenesis assays, virulence factors, transcription factors, effectors

## Abstract

The emergence of new fungal pathogens threatens sustainable crop production worldwide. One mechanism by which new pathogens may arise is hybridization. To investigate hybridization, the related smut fungi*, Ustilago maydis* and *Sporisorium reilianum*, were selected because they both infect *Zea mays*, can hybridize, and tools are available for their analysis. The hybrid dikaryons of these fungi grew as filaments on plates but their colonization and virulence in *Z. mays* were reduced compared to the parental dikaryons. The anthocyanin induction caused by the hybrid dikaryon infections was distinct, suggesting its interaction with the host was different from that of the parental dikaryons. Selected virulence genes previously characterized in *U. maydis* and their predicted *S. reilianum* orthologs had altered transcript levels during hybrid infection of *Z. mays*. The downregulated *U. maydis* effectors, *tin2, pit2,* and *cce1,* and transcription factors, *rbf1*, *hdp2*, and *nlt1,* were constitutively expressed in the hybrid. Little impact was observed with increased effector expression; however, increased expression of *rbf1* and *hdp2,* which regulate early pathogenic development by *U. maydis*, increased the hybrid’s capacity to induce symptoms including the rare induction of small leaf tumors. These results establish a base for investigating molecular aspects of smut fungal hybrid pathogen emergence.

## 1. Introduction

A major threat to food sustainability is the expanding virulence of existing pathogens and the emergence of new ones. In crop production, fungi are the most devastating group of pathogens [1,2], and hybridization is considered a major driver of their speciation. Hybridization is a process in which two individuals, from different populations that are distinguishable based on one or more heritable traits, either mate together or undergo vegetative fusion [3]. Most hybrids form transient interactions in nature that rarely lead to a speciation event [4,5,6]. However, genome sequencing has expanded the number of fungal species recognized as having hybrid origins. The hybridization process may be facilitated by the flexible genomes of fungi and the application of high selection pressures on pathogens in crop agriculture [7,8,9,10,11]. As an example, the devastating wheat stem rust, *Puccinia graminis* f. sp. *tritici* Ug99, which was recognized as a new pathotype with the capacity to infect 90% of the world’s wheat varieties, has recently been revealed to be a hybrid [12,13]. Genome sequencing and haplotype-resolved genome assembly followed by comparative analyses indicated that Ug99 emerged through somatic hybridization. This example indicates the potential impact of hybridization and supports the need to investigate the events of hybridization more thoroughly as it relates to fungal pathogenesis.

The Ustilaginaceae (basidiomycota) are a group of fungal pathogens that form biotrophic infections primarily targeting grasses of the Poaceae family [14,15]. These smut fungi are generally considered heterothallic and their pathogenicity is often tightly linked with their sexual cycle [16]. Sexual compatibility in most basidiomycetes is determined by two functionally conserved mating type loci, *a* and *b*, that encode a pheromone/pheromone receptor (PR) system and heterodimerizing homeodomain transcription factors, respectively [17,18,19,20]. In a study by Kellner et al. [21], sexual compatibility between various species within the Ustilaginaceae was assessed on plates. Induced filamentation by different combinations of smut haploids revealed extensive sexual compatibility within this group of fungi [21]. Following the discovery that *Ustilago bromivora* and *Ustilago hordei* were sexually compatible*,* comparative genomic analyses revealed that these two species were more closely related to each other than to *Ustilago maydis, Sporisorium reilianum,* and *Sporisorium scitamineum* [22]. Both fungi exhibit a dimorphic life cycle and a similar bipolar mating system, in which the *a* and *b* mating type loci reside on the same chromosome. *U. bromivora* and *U. hordei* infect *Brachypodium* sp. and barley, respectively, with *U. hordei* unable to infect *Brachypodium* sp. [23]. The hybridization of these two fungi formed stable dikaryons that could complete their life cycles to produce viable progeny upon infection of *Brachypodium* sp. The infections resulted in similar disease symptoms to those of *U. bromivora* with the authors suggesting that the hybrid may be more virulent based on the increased distorted spikelets and the presence of spores in the nodes of the stem. The Kellner et al. [21] study also indicated that the tetrapolar *U. maydis* and *S. reilianum* smuts, which are both known to target *Zea mays*, could fuse and form filaments on plates.

The known capability for fusion among strains with compatible mating type loci, and the existence of a common host for infection studies, provided us with the basis to further investigate the hybridization of *U. maydis* and *S. reilianum*. Selection of these species was also influenced by the discovery of gene expression pathways leading to pathogenesis, and functional analysis of virulence genes in *U. maydis* [24,25]. It was further supported by investigations of *S. reilianum* pathogenesis, and the availability of sequenced and annotated genomes for both species [26,27]. A significant distinction between the two fungi is the unique ability of *U. maydis* to infect and locally induce tumor formation on all aerial parts of maize [28,29] whereas *S. reilianum,* like other grass smuts, grows systemically, stimulating tumor formation only in the inflorescences [30]. *U. maydis* has been far more extensively studied than *S. reilianum* because of its rapidly induced symptoms and available molecular tools [26,31]. The research presented on the hybrid of *U. maydis* and *S. reilianum* including the creation of modified hybrids is discussed from the *U. maydis* perspective. The knowledge gained contributes to the understanding of fungal hybridization and identifies molecular changes that could influence pathogen emergence.

## 2. Materials and Methods

### 2.1. Strains and Growth Conditions

Fungal strains used in this study are listed in Table 1. *U. maydis* and *S. reilianum* strains were grown on either potato dextrose broth (PDB; 2.5% *w*/*v* potato dextrose broth; BioShop, Burlington ON, Canada) or yeast extract peptone sucrose (YEPS; 1% *w*/*v* yeast extract, 2% *w*/*v* peptone, 2% *w*/*v* sucrose; BioShop). On plates supplemented with 2% *w*/*v* agar (Sunrise Science Products, Knoxville TN, USA), *U. maydis* was grown for two days and *S. reilianum* for three days at 28 °C. Individual *U. maydis* and *S. reilianum* colonies were transferred from the plates to liquid and incubated for 18 and 24 h, respectively, at 28 °C, 250 rpm. All *U. maydis* expression mutants were grown at 28 °C in medium supplemented with 4 μg mL^−1^ carboxin (carb; BioShop) for three days on plates and 18 h in liquid. DH5α *Escherichia coli* strains were grown on LB (2.5% *w/v* Luria broth; EMD Millipore, Oakville ON, Canada) supplemented with 100 μg mL^−1^ ampicillin (amp; BioShop) and 2% *w*/*v* agar at 37 °C for 16–18 h. Colonies were inoculated into LB liquid with a 100 μg mL^−1^ amp and incubated at 250 rpm for 16–18 h. All fungal and bacterial strains were stored permanently in 15% *v*/*v* glycerol at −80 °C.

### 2.2. Mating Assays and Filamentous Growth

*U. maydis* and *S. reilianum* strains were grown in 3 mL PDB following the procedures indicated earlier. Cells were pelleted by centrifugation at 5000 rpm for 1 min, liquid was aspirated, cells washed with 1.0 mL sterile dH_2_O, centrifugation was repeated, and liquid was aspirated once more. Pellets were resuspended in dH_2_O to a final optical density (OD_600_) of 1.0. Equal amounts of each normalized haploid culture were combined and 5 µL was spotted on potato dextrose agar (PDA) plates supplemented with 1% *w*/*v* charcoal. Plates were stored at room temperature (RT) in the dark for three days. Filamentous growth was visualized daily using the Geliance 600 Imaging System (Perkin Elmer) and Leica S8 APO microscope (Leica Microsystems). Forced dikaryons and haploids were also harvested and frozen with liquid nitrogen (LN_2_) then stored at −80 °C for subsequent RNA isolation.

### 2.3. Seedling Pathogenesis Assays: Virulence Assessment, Quantification of Anthocyanin Coverage, and Time-Course Harvesting

*Zea mays* cv. Golden Bantam seeds (Ontario Seed Company, Kitchener ON, Canada) were planted (15 per 20 cm pot) in Sunshine Professional Growing Mix (Mix #1, Sun Gro Horticulture, Vancouver, BC, Canada). Seed germination and seedling growth were carried out in a Conviron CMP4030 growth chamber (18 h light, 70% RH, 30 °C; 6 h dark, 60% RH, 25 °C). For seedling infections, fungal strains were grown in 3 mL YEPS liquid before 50 μL of culture was transferred to 50 mL YEPS and incubated as described before. Culture concentrations were normalized to 1.0 OD_600_ with sterile dH_2_O. Equal volumes of compatible haploid cells were combined, and 0.5 mL was injected into the base of the seven-day old seedling using a 3 mL syringe and 18-gauge needle as described in Morrison et al. [34]. After infection, seedlings were fertilized every other day with Miracle-Gro All Purpose Plant Food 24-8-16 (80 g L^−1^).

Disease symptoms were scored at 5, 7, 10, 14, and 21 days post-infection (DPI), using the 0–8 scoring scale for *U. maydis* from Gold et al. [35] as amended by Goulet et al. [36]. The score of each plant was used to calculate an average, which was reported as the disease index (DI). The infection rate (IR) was measured by the percentage of plants displaying symptoms including the earliest indicator of fungal-plant interaction, chlorosis. Plant leaves were photographed, and the area of the total leaf and anthocyanin were measured, using ImageJ software version 1.53a, at 7 and 14 DPI [37]. A total of 10–45 infected leaves were processed from all three independent experiments. Plant tissue at 1, 3, and 7 DPI was harvested 0.5 cm above and below infection holes and consisted of leaf and stem tissue. At 7 DPI pure symptomatic leaf tissue was additionally harvested. Approximately 1.0 g of plant tissue was frozen in LN_2_ and stored at −80 °C until RNA isolation.

### 2.4. Microscopy

Cross sections of infected plant material were prepared using double-edged uncoated razor blades (American Safety Razor Company, Verona VA, USA) and transferred to a petri dish containing dH_2_O. Fungal filaments from mating assays and plant cross sections were separately mounted onto VWR VistaVision glass microscope slides (75 × 25 × 1 mm) and covered with a 1-ounce VWR Microcover glass coverslip (18 × 18 mm). Filaments were stained with ProLong Diamond Antifade Mountant with DAPI (4′6′-diamidino-2-phenylindole; Invitrogen, Mississauga ON, Canada), sealed, and incubated in the dark for ~16 h at RT. Cross sections were stained using the Fungi-Fluor Kit for Fungal Detection (Polysciences Inc, Warrington PA, USA). Microscopic observations were visualized using the Zeiss Axio Scope.A1 microscope at 100× and 400× magnification. Using the ZEN 2011 software (Carl Zeiss AG) photographs were taken with the AxioCam ICm1 (black and white). Unstained samples were observed with differential interference contrast (DIC) lighting and fluorescence of stained samples were observed under ultraviolet light (UV) excitation.

### 2.5. Accession Numbers, Orthology Assessment, and Sequence Analyses

The gene sequences were retrieved from the Fungi Ensembl database under the following accession numbers:

*Ustilago maydis: gapdh (UMAG_02491) KIS69978, rbf1 (UMAG_03172) KIS68601, hdp2 (UMAG_04928) KIS67055, nlt1 (UMAG_04778) KIS66714, tin2 (UMAG_05302) KIS66302, pit2 (UMAG_01375) KIS71480, cce1 (UMAG_12197) KIS68887*.

These previously characterized *U. maydis* genes were used in reciprocal best hit (RBH) protein blast (BlastP) searches to identify potential *S. reilianum* orthologs using the criteria that the most likely orthologs are those genes that were the top hit of one another in reciprocal searches. The *S. reilianum* genes, distinguished here by having an ‘*sr*’ prefix, were identified and retrieved for analysis under the accession numbers:

*Sporisorium reilianum: srgapdh (sr10940.2) CBQ71234, srrbf1 (sr14215) CBQ73544, srhdp2 (sr15806) CBQ69341, srnlt1 (sr15650) CBQ69731, srtin2 (sr10057) CBQ70078, srpit2 (sr10529) CBQ70785, srcce1 (sr13927) CBQ73267*.

The amino acid sequences of *U. maydis* and *S. reilianum* orthologs were aligned using MAFFT [38] and visualized in Jalview with a ClustalX color scheme [39]. Domains were identified through PROSITE [40] and PFAM [41] and signal peptides identified using SignalP [42].

### 2.6. Total RNA Isolation, DNaseI Treatment, Reverse Transcription, and Species-Specific Transcript Analysis

Mating assay mycelia and ~1.0 g samples of infected plant tissue were ground in LN_2_ with a sterile mortar and pestle prior to RNA isolation. Axenic haploid strains were grown in 3 mL YEPS before 50 μL of culture was transferred to 50 mL YEPS liquid and incubated as described before. Cells were pelleted once they reached an OD_600_ of 1.0 and the supernatant was removed. Total RNA was separately extracted from harvested mycelia, ground plant tissue, and pelleted cells using TRIzol reagent, precipitation, treatment with DNaseI, and subsequent check for genomic DNA contamination was performed as described in Zahiri et al. [43] and updated by Morrison et al. [34]. Once genomic contamination was removed, the treated RNA was diluted to either 100 or 150 ng µL^−1^, then 400–1200 ng of template was used for first-strand synthesis reactions. Reverse transcription was carried out using the TaqMan Gold RT-PCR Kit (Applied Biosystems, Connecticut MA, USA) in 20 µL reaction volumes, according to the conditions listed in Morrison et al. [34]. Resulting cDNA was diluted with sterile dH_2_O (1:3) before use as PCR template.

Each reverse transcriptase-polymerase chain reaction (RT-PCR) contained 13.375 µL dH_2_O, 2.5 µL 10× Gold Buffer, 3 µL 25 mM MgCl_2_, 2 µL 2.5 mM dNTPs, 0.125 µL 5 U µL^−1^ AmpliTaq Gold, 1 µL 5 µL of each species-specific primer, and 2 µL cDNA template. When cDNA template concentrations were low, 4 µL of template was used with the volume of dH_2_O adjusted. RT-PCR was performed in a Veriti thermocycler (Applied Biosystems) using the following conditions: 95 °C for 10 min, 35 cycles of (95 °C for 30 s, X °C for 30 s and 72 °C for 30 s), 72 °C for 10 min then held at 4 °C. X °C refers to the primer annealing temperature as indicated in Appendix A. One third of the RT-PCR product was electrophoretically separated on a 1.2% agarose gel (1× TAE) at 90 V for 1 h and visualized by ethidium bromide staining (0.3 µg mL^−1^, EtBr; BioShop). Sizes were compared to FullRanger DNA ladder (Norgen Biotek, Thorold ON, Canada).

RT-qPCR analysis used Power SYBR Green PCR Master Mix (Applied Biosystems) and species-specific primers (Appendix A). Reactions contained 4 µL dH_2_O, 2 µL 5 µL of each primer, 10 µL Power SYBR Green PCR Master Mix, and 2 µL cDNA template. RT-qPCR was performed in a QuantStudio 3 (Applied Biosystems) using the following conditions: hold stage (50 °C for 2 min then 95 °C for 10 min), PCR stage 40 cycles of (95 °C for 15 s, X °C for 1 min), and melt curve stage (95 °C for 15 s, 60 °C for 1 min, 95 °C for 15 s). X °C refers to the annealing temperature indicated in Appendix A. Technical replicates were run in triplicate.

For RT-PCR and RT-qPCR, gene transcripts were normalized against the housekeeping gene encoding glyceraldehyde-3-phosphate dehydrogenase (*gapdh*), and axenically grown non-pathogenic haploids were used as a transcript level reference for comparison. For RT-qPCR, relative expression values were calculated using the 2^−^^ΔΔCt^ method of Livak and Schmittgen [44].

### 2.7. Statistical Analyses

Significant differences in strain virulence assessed by seedling pathogenesis assays were calculated using non-parametric Mann–Whitney U tests. Student’s *t*-test with a Welch correction was used to calculate the significant differences in the percentage of anthocyanin induced on the leaf surface area. Finally, significant differences in transcript levels, detected by RT-qPCR, between 521 × SRZ2 and each parental dikaryon during time-courses of pathogenic development were assessed using Student’s multiple paired *t*-test with a Welch correction. All statistical analyses were completed using Prism 8 software (GraphPad, San Diego CA, USA).

### 2.8. Creation of U. maydis Expression Constructs and Strains

Six gene constructs were created to be ectopically expressed from the *ip* (iron-sulphur) locus of the *U. maydis* strain, 521, and creation of these strains conferred carboxin resistance. In the p123 shuttle vector containing a constitutive promoter (P_otef_), the open reading frame (ORF) of each *U. maydis* gene replaced the green fluorescent protein (GFP) of p123. Two expression constructs, p123 + P_otef_:*tin2* and p123 + P_otef_:*nlt1,* were generously provided by Jessie Fetsch and Justin Meade, respectively. For the creation of the other four expression constructs, manufacturer’s protocols were followed for each indicated kit unless stated otherwise. To amplify each ORF, cDNA template and primers with restriction endonuclease (RE) cleavage sites at the 5′ and 3′ ends (Appendix A) were used. Reactions amplifying *rbf1* and *cce1* ORFs used Phusion High Fidelity DNA Polymerase (Thermo Scientific, Mississauga ON, Canada) while Elongase Enzyme Mix (Invitrogen) was used to amplify *pit2* ORF. All PCR products were purified using the PureLink PCR Purification Kit (Invitrogen). The ORF of *hdp2* contained the *Not*I site necessary for cloning into the p123 vector, therefore *hdp2* was synthesized without the *Not*1 site and provided in the plasmid vector pUC57 (GenScript, Piscataway NJ, USA). In the synthesized *hdp2* DNA, the sequence GGCCGC was replaced with GGCAGC to retain the same amino acid, alanine, with the *Not*I recognition site removed. The plasmid DNA (pDNA) was prepared and diluted with sterile dH_2_O to 5 ng μL^−1^. The diluted pUC57 + *hdp2* DNA was transformed into Subcloning Efficiency DH5α Competent Cells (Invitrogen). Both p123 and pUC57 + *hdp2* pDNA were isolated from *E. coli* culture using PureLink Quick Plasmid MiniPrep kit (Invitrogen). To create compatible ends for ligation, *Not*I-HF and *Bam*HI-HF (New England BioLabs, Whitby ON, Canada) were used to digest p123 and the amplicons of *hdp2,* and *pit2* whereas *Not*I-HF and *Nco*I-HF (New England BioLabs) were used to digest p123 and the amplicons of *rbf1,* and *cce1*. To obtain pure vector backbone and ORF, the digested fragments from p123 and *hdp2* ORF were gel extracted using the PureLink Quick Gel Extraction Kit (Invitrogen) and the remaining ORFs were PCR purified using the PureLink PCR Purification Kit (Invitrogen). The p123 backbone and ORF were ligated with T4 DNA ligase (New England Biolabs) and transformed into Subcloning Efficiency DH5α Competent Cells (Invitrogen), followed by a pDNA isolation as indicated above. For sequencing each construct, reactions were set up as described in Ho et al. [45] (Appendix A) using the Big Dye v3.1 Cycle Sequencing Kit (Thermo Scientific, Mississauga ON). The reaction products were separated on an ABI 3730 DNA analyzer. Sequences were analyzed using Sequence Scanner Software 2.0 (Applied BioSystems), SeqMan Pro^TM^ 11.2.1 (DNASTAR Inc., Madison WI, USA), and MEGA 7.0.26.

To prepare competent *U. maydis* protoplasts, Strain 521 was grown in 100 mL complete medium (CM; 0.25% *w*/*v* casamino acids, 0.15% *w*/*v* anhydrous ammonium acetate, 1% *w*/*v* yeast extract, 1% *w*/*v* sucrose, 6.25% *v*/*v* salt solution- see Ho et al. [46]) to 1.5 OD_600_ before following the protocol outlined by Wang et al. [47] and modified by Ostrowski and Saville [48]. Prior to transformation, 10–15 μg of pDNA was linearized using *Ssp*I-HF (New England BioLabs) and purified using the PureLink PCR Purification Kit (Invitrogen). Linearized pDNA was transformed into competent 521 protoplasts following the protocol of Yee [49] as modified by Morrison et al. [34]. Genomic DNA was isolated from the culture using an organic extraction described by Hoffman and Winston [50]. PCR screens were used to test each generated mutant before confirmation by Southern hybridization using the DIG High Prime DNA Labeling and Detection Starter Kit 1 (Roche Diagnostics, Laval QC, Canada) with a DIG-labelled probe for the carboxin gene.

## 3. Results

### 3.1. Induced Filamentation and Virulence Assessment of U. maydis and S. reilianum Hybrids Revealed Sexual Compatibility but Reduced Virulence during Infection

Compatibility of haploid cultures was assessed by co-spotting on PDA + charcoal. The *U. maydis* forced dikaryon, 518 × 521, produced white fuzzy filamentous growth while the *S. reilianum* forced dikaryon, SRZ1 × SRZ2, produced limited filamentous growth (Figure 1A). The hybrid, 521 × SRZ2, produced filamentous growth indicating compatibility while 518 × SRZ1 did not, consistent with Kellner et al. [21]. Close monitoring over three days revealed that the white fuzzy filaments of 521 × SRZ2 at three days were comparable to that of 518 × 521 at two days (Appendix A). Filaments of 518 × 521 and 521 × SRZ2 were DAPI stained revealing that the mycelia contained two nuclei consistent with dikaryon formation (Figure 1B). Since SRZ1 × SRZ2 showed limited filamentation on PDA + charcoal, it could not be successfully DAPI stained.

To assess the virulence of the *U. maydis* × *S. reilianum* hybrids, seedling pathogenesis assays were conducted, using the *Z. mays* cv. Golden Bantam unless stated otherwise. Virulence was assessed as the disease index (DI) and infection rate (IR) for 21 days post infection (DPI) with 7 and/or 14 DPI reported here. Using the same strain combinations as the plate assays, an initial seedling infection assay showed high infection rates and virulence of the parental dikaryons (Appendix A). *U. maydis* infections induced localized symptoms such as chlorosis, anthocyanin production, and tumor formation on the leaf and stem. In contrast, *S. reilianum* infections were systemic and only induced anthocyanin production. The 521 × SRZ2 hybrid combination infected maize at a lower rate than the parental dikaryons, and its symptoms of infection were only chlorosis and anthocyanin indicating a lower virulence relative to *U. maydis* infections. The only characteristic of the 518 × SRZ1 infections was minor chlorosis in a limited number of plants comparable to the infections using haploid strains.

Focusing on the 521 × SRZ2 infections, with the observed slower development of the hybrid dikaryon filaments on plates it was considered possible that its development in *Z. mays* was also delayed. If this was the case, then by the time the hybrid infects the seedling, following inoculation, the seedling may be older than those infected by *U. maydis*. To determine if the age of the seedlings had an impact on the hybrid dikaryon’s ability to infect, we inoculated 6, 7, 8 and 9-day old seedlings with 521 × SRZ2. At 14 DPI, the infection rate of 521 × SRZ2 was significantly reduced in the 9-day old seedling inoculations relative to the 6 and 7-day old seedling inoculations (Appendix A). There was no significant difference between the 6 and 7-day old seedlings infected with 521 × SRZ2. As a control, *Z. mays* seedlings were also separately infected with 518 × 521. A lower DI was found in the 8 and 9-day old seedling inoculations compared to that of the 6 and 7-day old seedlings inoculations at 5 and 7 DPI. However, no significant difference was found in the *U. maydis* infection rates and symptoms when comparing the infections of 6, 7, 8 and 9-day old seedlings at 14 DPI.

Inoculations of *Z. mays* cv. Golden Bantam cobs, as described by Zahiri et al. [43], demonstrated that while 518 × 521 produced tumors, 521 × SRZ2 did not (data not shown). To assess if 521 × SRZ2 grew systemically and was able to induce tumors in the inflorescences, seven-day old *Z. mays* cv. Golden Bantam, Gaspe Flint, B73, and Painted Mountain were separately inoculated with 521 × SRZ2 and SRZ1 × SRZ2. Symptoms were continually monitored until after the formation of *Z. mays* inflorescences with SRZ1 × SRZ2 inducing tumor formation while 521 × SRZ2 did not (data not shown). Subsequent *Z. mays* infection assays were completed using standard assay conditions with seven-day old seedlings.

The symptoms of seedling pathogenesis assays were scored from 5 to 21 DPI (Figure 2 and Appendix A). The infections with 518 × 521 at 7 DPI had a DI of 3.52, in which leaf tumors were the main symptom. At 14 DPI, symptom severity had increased with the formation of larger leaf tumors and the presence of stem tumors with approximately 20% of plants being dead (Figure 2A,B). The SRZ1 × SRZ2 infections produced minor symptoms of chlorosis and anthocyanin induction at 7 and 14 DPI with a DI of 1.40 and 1.90, respectively. By 14 DPI, both parental dikaryon infections had an IR above 90% while 521 × SRZ2 infections resulted in a significantly lower IR of 70.4%. The infection of 521 × SRZ2 produced only symptoms of chlorosis and anthocyanin with a DI of 0.83 and 1.40 at 7 and 14 DPI, respectively. The anthocyanin induced by the infections differed in pigmentation, those of *U. maydis* induced a red-orange color, *S. reilianum* a dark red-purple color mixed with chlorotic patches, and the hybrid induced primarily solid patches of dark red-purple, similar in color to that induced by *S. reilianum* but without any chlorotic spots (Figure 2B). The percentage of anthocyanin induced by the hybrid infections on the leaf surface area at 7 DPI was approximately 25%, which was similar to each parental dikaryon (Figure 2C). The leaf area in which anthocyanin was induced by the hybrid at 14 DPI was significantly smaller than that of the parental dikaryon infections.

Cross sections at different timepoints of 518 × 521, 521 × SRZ2, and SRZ1 × SRZ2 infections were stained with Fungi-Fluor. At 5, 7, 10, 14, and 21 DPI, 15–30 fields of view at 100× magnification were examined for hyphal growth during each dikaryon infection. Due to the limited fungal proliferation, hyphal growth was not quantified, but visual observations revealed that at 5 DPI, 518 × 521, 521 × SRZ2, and SRZ1 × SRZ2 infections had a low level of fungal colonization (Appendix A). At later time points, 518 × 521 infections showed extensive fungal proliferation, consistent with the increased disease symptoms observed (Figure 3 and Appendix A). At 7 DPI, 521 × SRZ2 and SRZ1 × SRZ2 infections still had relatively low fungal proliferation. At the later timepoints, SRZ1 × SRZ2 began to extensively proliferate, yet remained lower than 518 × 521, while the 521 × SRZ2 infections continued to have low hyphal growth at all timepoints. Within the *Z. mays* cells, branch primordia and branching were seen on the hyphae of all three dikaryon infections.

### 3.2. Reduced Transcript Levels in the Hybrid Dikaryon during Pathogenic Development Revealed Genes to Constitutively Express in the Hybrid

To gain insight regarding the reduced symptoms and hyphal growth resulting from hybrid infections, transcript levels of virulence genes were investigated. For this, virulence genes encoding transcription factors, effectors, and other cell signaling proteins were selected (Appendix A). The majority of these genes, when deleted in *U. maydis* by our lab and other research groups, showed a reduction in virulence indicating their important role in pathogenic development. Another consideration in gene selection was the timing of their function with a focus on early penetration and fungal proliferation in the host since the hybrid seems to progress to an early stage and then stall when compared to *U. maydis* infections. For each gene, the *S. reilianum* ortholog was identified using reciprocal best hit protein blast. RT-PCRs with species-specific primers were used to screen for transcript level changes in the RNA isolated from mycelia harvested from PDA + charcoal plates and *in planta* pathogenic time-courses for 518 × 521, 521 × SRZ2, and SRZ1 × SRZ2 relative to the axenically grown non-pathogenic haploid strains (Appendix A). In these screens, we identified gene transcript levels that were reduced or not detected in the hybrid dikaryon when compared to the parental dikaryons in the pathogenic time-courses. In selecting genes for further investigation, we considered the gene’s function (Appendix A) and the timing of its activity during pathogenic development in *U. maydis* according to Lanver et al. [24]. Six genes were selected for constitutive expression in *U. maydis* strain 521 consisting of three early wave effectors, *tin2, pit2,* and *cce1* and three transcription factors, *rbf1, hdp2,* and *nlt1,* which are involved in different stages of *U. maydis* pathogenic development.

The *U. maydis* and *S. reilianum* effector orthologs *tin2* and *pit2* had percent identities of 41.8% and 33.7%, respectively, while *cce1* putative orthologs had 63.9% identity (Appendix A). Amino acid alignments of the putative effector orthologs revealed that the signal peptides were not conserved (Appendix A). The amino acid sequence similarity among the selected transcription factors was higher than that of the effectors. The *U. maydis* and *S. reilianum rbf1* orthologs had the highest identity at 69.9% while *hdp2* and *nlt1* had identities of 57.7% and 61.6%, respectively (Appendix A). The amino acid and functional domain alignments revealed conservation of their DNA binding domains (Appendix A). Species-specific primers were also designed for the effectors and transcription factors then synthesized for RT-qPCR to verify the RT-PCR screens. The transcript levels of these genes were assessed during pathogenic time-courses and calculated relative to the transcript levels in axenically grown non-pathogenic haploid strains.

During 518 × 521 infections, *tin2* transcript levels were upregulated at all timepoints while only being detected in the pure leaf tissue of 521 × SRZ2 infections (Figure 4). The transcripts of *srtin2* showed similar levels in both SRZ1 × SRZ2 and 521 × SRZ2 infections. In 518 × 521 infections, *pit2* was substantially upregulated at all time points in both mixed tissue and pure symptomatic leaf tissue. This gene was also upregulated in the 521 × SRZ2 infections, but at significantly lower levels than that of 518 × 521 except in the pure symptomatic leaf tissue where the levels were comparable. SRZ1 × SRZ2 and 521 × SRZ2 had comparable levels of *srpit2* at all stages of infection and tissue types. During 518 × 521 infection, *cce1* was upregulated at all time points and tissue types in a similar pattern to *pit2* but at lower levels. Within 521 × SRZ2 *cce1* transcripts were detected at lower levels than those in 518 × 521 but the differences were only significant at 3 DPI. Both *cce1* and *srcce1* were comparably upregulated within 521 × SRZ2 at all time points and tissue types except 1 DPI when *srcce1* was expressed at a lower level. Within SRZ1 × SRZ2, *srcce1* was upregulated to a greater extent than in 521 × SRZ2 except in pure leaf tissue where the levels were comparable.

Transcript analysis revealed *rbf1* transcripts were elevated following similar patterns in 518 × 521 and 521 × SRZ2, although the latter had slightly lower levels at 1 and 3 DPI (Figure 4). Transcripts of *srrbf1* were only detected in the 521 × SRZ2 at 3 and 7 DPI in mixed tissue whereas as they were detected in SRZ1 × SRZ2 at all timepoints and tissue types. In 518 × 521, *hdp2* was expressed at a high level during all time points and in both tissue types while in the 521 × SRZ2 infections this gene was detected only at 1 DPI in mixed tissue and at a lower level. However, the *srhdp2* transcript levels were comparable between SRZ1 × SRZ2 and 521 × SRZ2 infections for all timepoints and tissue types. In both 518 × 521 and SRZ1 × SRZ2, *nlt1* was detected at 3 and 7 DPI with similarly high levels, except in the pure leaf tissue where *nlt1* was much lower in SRZ1 × SRZ2. In contrast, transcripts of *nlt1* and *srnlt1* were undetected in 521 × SRZ2.

### 3.3. Constitutively Expressed Virulence Genes within the Hybrid Revealed Transcription Factors Were Capable of Influencing Pathogenic Development

To assess if the stall in the hybrid’s pathogenic development could be overridden, previously characterized *U. maydis* effectors *tin2*, *pit2,* and *cce1* as well as transcription factors, *rbf1*, *hdp2,* and *nlt1* were constitutively expressed within the 521 haploids of the hybrid. Mating assays were carried out to visually assess the impact of constitutively expressing these genes on the formation of the filamentous dikaryons. No observable difference in filament formation was found in 521 × SRZ2 or 518 × 521 fusions when *tin2, pit2,* and *cce1* were constitutively expressed. Similar filamentation was induced by 521 × SRZ2 and 521 P_otef_:*rbf1*^SI^ × SRZ2 while 521 P_otef_:*rbf1*^MI^ × SRZ2 had slightly increased filamentation (Figure 5). The haploid, 521 P_otef_:*rbf1*^MI^, induced minor filamentation on its own which confounded the interpretation of filamentation appearing in its co-spotting with the previously non-compatible SRZ1. The single and multiple inserts of 521 P_otef_:*hdp2* × SRZ2 and 521 P_otef_:*nlt1* × SRZ2 had pronounced reduction in filamentation relative to 521 × SRZ2. A reduction, although not to the same extent, was also observed during the *U. maydis* fusions of the 521 P_otef_:*hdp2* and 521 P_otef_:*nlt1* strains with the compatible haploid, 518. Despite the reduced filamentation in the dikaryons, the 521 haploid expressing *hdp2* from a single and multiple insert also induced minor filamentation but less than what was observed in haploid in which *rbf1* was expressed.

Seedling pathogenesis assays were conducted to determine if the constitutively expressed effectors influenced the pathogenic development of the hybrid. Virulence of 521 P_otef_:*tin2*^SI^ × SRZ2 and 521 P_otef_:*tin2*^MI^ × SRZ2 was not different from that of 521 × SRZ2 inoculations at any timepoint; however, the infection rate was slightly reduced at 7, 10, and 14 DPI (Appendix A). Seedling pathogenesis assays of 521 P_otef_:*pit2*^SI^ × SRZ2 and 521 P_otef_:*pit2*^MI^ × SRZ2 revealed no significant difference compared to 521 × SRZ2 infections at any timepoints (Appendix A). The only notable result at 7, 10, and 14 DPI was that the infection rate of 521 P_otef_:*pit2*^SI^ × SRZ2 was 10% lower than 521 × SRZ2 yet it was 10% higher in 521 P_otef_:*pit2*^MI^ × SRZ2. This led to a significant difference between 521 P_otef_:*pit2*^SI^ × SRZ2 and 521 P_otef_:*pit2*^MI^ × SRZ2 infections at 7 (*p* = 0.0023), 10 (*p* = 0.0014), and 14 (*p* = 0.0316) DPI. Infections with 521 P_otef_:*cce1*^SI^ × SRZ2 and 521 P_otef_:*cce1*^MI^ × SRZ2 were not significantly different than 521 × SRZ2 (Appendix A). At 10 and 14 DPI, there was a trend toward increased infection rate proportional to the number of inserts of *cce1*, although no significant differences were detected. As observed for the other two effectors, there was no notable phenotypic differences observed nor impact on virulence.

Seedling pathogenesis assays were also used to assess the influence of constitutive transcription of genes encoding the transcription factors *rbf1*, *hdp2,* and *nlt1* on hybrid pathogenesis. At 7 DPI, 521 P_otef_:*rbf1*^MI^ × SRZ2 had a higher IR (84.4%) compared to 521 P_otef_:*rbf1*^SI^ × SRZ2 (57.0%) or 521 × SRZ2 (54.1%) (Figure 6A). Infections with the hybrid dikaryon expressing multiple inserts of *rbf1* also induced anthocyanin mixed with chlorotic patches (Figure 6B). The infection rate of 521 P_otef_:*rbf1*^SI^ × SRZ2 became statistically different than 521 × SRZ2 at 10 DPI but remained significantly lower than 521 P_otef_:*rbf1*^MI^ × SRZ2 throughout the assay (Appendix A and Figure 7A). The coverage of anthocyanin on the leaf surface was significantly greater in both hybrids expressing *rbf1* relative to 521 × SRZ2 at 7 and 14 DPI (Figure 7C). Anthocyanin induction remained the primary symptom but in a rare event, small leaf tumors were identified in the 521 P_otef_:*rbf1*^MI^ × SRZ2 infection at 14 DPI (Figure 7A,D). Approximately 5–8 cross sections of 521 P_otef_:*rbf1*^SI^ × SRZ2 and 521 P_otef_:*rbf1*^MI^ × SRZ2 infected *Z. mays* that displayed symptoms showed similar level of hyphal growth to 521 × SRZ2 infections at 7 and 14 DPI (data not shown).

At 7 DPI, 521 P_otef_:*hdp2*^SI^ × SRZ2 and 521 P_otef_:*hdp2*^MI^ × SRZ2 assay results were significantly different from 521 × SRZ2 (Figure 7A). The 38.5% IR of 521 × SRZ2 was low compared to the 83.7% and 81.3% IR of 521 P_otef_:*hdp2*^SI^ × SRZ2 and 521 P_otef_:*hdp2*^MI^ × SRZ2, respectively. By 10 and 14 DPI, about 90% of plants displayed anthocyanin when infected with either 521 P_otef_:*hdp2*^SI^ × SRZ2 or 521 P_otef_:*hdp2*^MI^ × SRZ2 and this was significantly higher than anthocyanin induction by 521 × SRZ2 (Figure 7A and Appendix A). For the hybrid dikaryons expressing single and multiple inserts of *hdp2*, the anthocyanin observed was mixed with chlorotic patches (Figure 7B). As observed with the *rbf1* constitutively expressed strains, there was significantly greater anthocyanin induction on the leaf surface in the strains constitutively expressing *hdp2* relative to infections with 521 × SRZ2 (Figure 7C). Small leaf tumors were observed on rare occasion with plants infected by both the single and multiple inserts of *hdp2* within the hybrid at 14 DPI (Figure 7A,D). Cross sections of 521 P_otef_:*hdp2*^SI^ × SRZ2 and 521 P_otef_:*hdp2*^MI^ × SRZ2 infected *Z. mays* revealed similar amounts of hyphal growth to 521 × SRZ2 infections (data not shown).

Seedling pathogenesis assays with hybrids constitutively expressing *nlt1* revealed 521 P_otef_:*nlt1*^SI^ × SRZ2 and 521 P_otef_:*nlt1*^MI^ × SRZ2 infections were significantly different than 521 × SRZ2 (Figure 8). This significance was due to a higher number of infected plants displaying distinct chlorosis that covered a larger area on the leaf surface. Despite higher infection rates when anthocyanin was induced the coverage was less pronounced than in 521 × SRZ2 infections. In the later timepoints, 521 × SRZ2 infections had a higher virulence rating as indicated by the number of plants displaying anthocyanin while the *nlt1* expression strains had higher IR because the presence of chlorosis was scored as infection (Appendix A).

Transcript levels were assessed for all six genes that were constitutively expressed as single and multiple inserts in 521 × SRZ2 during infections at 3 and 7 DPI (Appendix A). They were also determined in the axenic haploid cultures constitutively expressing each gene. Transcripts levels were proportional to the number of inserts of each expressed gene and were higher during infection compared to the axenically grown cultures. Two notable exceptions include the constitutively expressed *pit2* single insert, which was expressed at a comparable level to the *pit2* in the 521 × SRZ2 infection at 3 DPI and then downregulated at 7 DPI and the constitutively expressed *hdp2* single and multiple inserts*,* which were only detected at 3 DPI. RT-qPCR data of 521 × SRZ2 were plotted with each altered dikaryon to provide context.

## 4. Discussion

The sexual cycle of fungal pathogens belonging to the Ustilaginaceae is tightly linked to their pathogenic development. The initial step in both processes is the fusion of haploids which requires compatibility at the mating type loci. *Ustilago maydis* and *Sporisorium reilianum* are among the smut fungi found to be capable of interspecies haploid cell fusion [21] and they have the same host, *Zea mays*. Kellner et al. [21] demonstrated that only *U. maydis* 521 (a1b1) and *S. reilianum* SRZ2 (a2b2) combination could induce filamentation on plates whereas combinations involving 518 (a2b2), SRZ1 (a1b1), and SRZCXI1 (a3b2) did not filament. Under our laboratory conditions mating assays confirmed that *U. maydis* 521 and *S. reilianum* SRZ2 can fuse to form a hybrid dikaryon as demonstrated by the formation of long, straight, unbranched filaments on plates, which indicates compatibility between both *a* and *b* loci [51]. These assays also revealed that the hybrid had morphologically similar filamentous growth to the *U. maydis* dikaryon but that it developed slower. This was also reflected during pathogenesis assays where visible signs of infection by the hybrid, notably the induction of anthocyanin production, were delayed relative to the *U. maydis* infections. While hybrid infections induced anthocyanin production, they did not induce tumor or teliospore formation in the seedlings or inflorescences, indicating that the later events of pathogenesis exhibited by either *U. maydis* or *S. reilianum* do not occur during hybrid infections. Stained cross sections of infected *Z. mays* seedlings revealed that the hybrid was able to penetrate the plant and enter plant cells, however, the extent of mycelial development following infection was substantially less than that of either parental dikaryon. The inability of the hybrid to further colonize the host cells could be the result of a stall in the fungal developmental pathway or that the interactions between the hybrid and the plant led to a block in development. RT-PCR screens and RT-qPCR analyses revealed transcript levels within the hybrid during infections were distinct from those in the *U. maydis* and *S. reilianum* dikaryon infections, which likely contributed to the alteration in the hybrid’s pathogenic development. Differences were also detected in the transcription between the two nuclei within the hybrid, which could be the outcome of each nucleus responding differently to infection because of distinct signaling or that transcription differed with the same signaling. Distinguishing between these possibilities was beyond the scope of this study.

Anthocyanin production, the primary phenotype of the hybrid dikaryon infections, differed in timing of induction, pigmentation, and the amount induced on leaf surface area when compared to the *U. maydis* and *S. reilianum* dikaryon infections of *Z. mays*. During *U. maydis* infections of *Z. mays*, anthocyanin induction can be observed as early as 2–3 DPI although more typically it is seen at 5–6 DPI [52]. Our assays were consistent with this, and we showed that *tin2* transcript levels were elevated at 1, 3, and 7 DPI which reflects the role this effector has in inducing anthocyanin production. In contrast, our seedling pathogenesis assays with *S. reilianum* induced strong anthocyanin production while others have noted that *S. reilianum* infections do not [53]. During our *S. reilianum* infections, seedlings that showed signs of anthocyanin developed tumors when the inflorescences were formed indicating that anthocyanin induction is an early symptom of *S. reilianum* infections of the corn variety used in our assays. The anthocyanin induced by *U. maydis, S. reilianum,* and the hybrid dikaryon infections differed in pigmentation as *U. maydis* induced a red-orange color and *S. reilianum* a dark red-purple color mixed with chlorotic patches while the hybrid induced primarily solid patches of dark red-purple. At 14 DPI, the area of the leaf surface with induced anthocyanin was significantly less in the hybrid infections than those of its parents. Production of anthocyanin has been identified as a plant response to various environmental stresses including photoinduction, temperature, nutrient stress, and pathogen infection as reviewed in Chalker-Scott [54]. Infection of *U. maydis, S. reilianum,* and hybrid dikaryons were completed at the same time therefore factors like soil, growth chamber conditions, and water + fertilizer schedule were consistent indicating environmental factors or stresses were unlikely to be a source of the differences observed. These seedling pathogenesis assays were also separately repeated three times and each time produced similar results. The differences found in the anthocyanin phenotype suggests that the hybrid’s initial interactions with the host are distinct from both parents. The basis for these differences between the hybrid and the parental dikaryons is not known but could be a result of each dikaryon interacting differently with the biosynthetic pathway for anthocyanin in the host.

The hybrid’s inability to induce symptoms beyond the induction of anthocyanin and complete its life cycle led us to consider the possibility that the parental haploids did not fuse to form the infectious dikaryon during infection of *Z. mays*. The data presented here suggest that plant inoculations with 521 × SRZ2 resulted in the formation of an infectious dikaryon and that pathogenic development was blocked after the induction of anthocyanin production. Anthocyanin production was seen in 521 × SRZ2 inoculations but not with haploid inoculations or inoculations with the sexually incompatible 518 × SRZ1. In *U. maydis* dikaryon infections anthocyanin induction indicates a biotrophic interaction has occurred with the host. The induction of anthocyanin by hybrid infection may similarly reveal the initiation of a biotrophic interaction but the pigmentation of anthocyanin production, timing of induction, and the amount on the leaf surface indicates a different interaction with *Z. mays*. The filamentous growth of the hybrid in the plant with the production of branch primordia are consistent with the production of a functional bE/bW protein heterodimer [55]. The presence of an active heterodimer is also supported by the induction of *U. maydis* downstream targets of bE/bW, *rbf1* and *hdp2*. However, the transcriptional cascade during *U. maydis* pathogenic development results in the expression of *nlt1* and transcripts of this gene were not detected during hybrid infections suggesting a block in the transcriptional cascade. Together, this data supports that 521 and SRZ2 cells fuse, dikaryons form and penetrate the plant, effectors are released inducing anthocyanin production, but that further pathogenic development is blocked or stalled.

The stall corresponded to reduced transcript levels of effectors and transcription factors with previously determined roles in the pathogenic development of *U. maydis*. We proposed that restoring the expression of these genes could enhance the ability of the hybrid to continue in its development toward tumor and teliospore formation. Expressing the end products of signaling, fungal effectors, may bypass the need for signal transduction and enable pathogenesis to proceed. Effectors often have small additive impacts on virulence; however, the *U. maydis tin2, pit2,* and *cce1,* were noted to have more substantial roles and were chosen for constitutive expression in the hybrid [56,57,58]. Another way to compensate for altered signaling between the *Z. mays* and the hybrid is to constitutively express transcription factors, that are the targets of signal transduction, and thus initiate cascades of gene expression without the need for signals. As such, three transcription factors, *rbf1, hdp2,* and *nlt1,* identified as main drivers of different stages of pathogenic development were selected to be overexpressed [24]. The direct impact of effectors on plant pathogen interactions suggests that any impact the transcription factors have would likely be mediated through these proteins.

The induction of anthocyanin production is the main symptom of infection by the hybrid. During *U. maydis* infections, the effector *tin2* is proposed to be responsible for this induction by redirecting biosynthesis away from lignin production and into anthocyanin synthesis [56]. This reduces the ability of the plant to strengthen cell walls making them more susceptible to fungal penetration. In this investigation, *tin2* transcripts were detected at all time points in *U. maydis* infections and anthocyanin production was induced. However, during hybrid infections *tin2* transcripts were only detected in pure leaf tissue at 7 DPI. If the role of Tin2 in the hybrid was the same as proposed for *U. maydis* then the delayed and reduced expression of *tin2* in the hybrid would result in the inability to reduce lignin biosynthesis, which may make it difficult for the hybrid to colonize *Z. mays*. However, the hybrid dikaryons constitutively expressing *tin2* had similar virulence and anthocyanin production to that of 521 × SRZ2. Together these data suggest that the anthocyanin observed in the hybrid was induced through a different mechanism than that proposed for *U. maydis* [53].

In *U. maydis,* the effectors, *pit2* and *cce1,* are involved in suppressing plant defenses in the apoplastic interface between the fungus and plant cells [58,59]. The low infection rate and lack of fungal proliferation inside the host is consistent with the hybrid’s inability to suppress the host defenses. Transcript levels of *pit2* and *cce1* as well as the *S. reilianum* orthologs in the hybrid were less than those of the parental dikaryons, however, the pattern of expression was similar. Constitutive expression of *pit2* had a minor impact on infection rate, but no other observable effect. This was surprising given its role in virulence in both parental dikaryons [57,60]. Constitutively expressing *cce1* in the hybrid did not alter virulence and minimally impacted the infection rate, despite evidence indicating *cce1* is capable of influencing virulence when expressed in different smut fungi [58]. While this data is limited to only three effectors, it suggests that altered expression of a single effector is not capable of altering the virulence of the 521 × SRZ2 hybrid.

While it may be necessary to express more than one effector or to have targeted expression of effectors to influence hybrid pathogen emergence, the secretion of the effectors may also play a role. The *U. maydis* and *S. reilianum* effectors used here have different signal peptides which could have influenced their secretion. Further there was a lack of extensive branching by the hybrid during infection which means the amount and/or location of secretion was likely reduced relative to that seen during infections by the parental dikaryons [61]. However, successful infection of *Brachypodium* sp. by *U. bromivora* × *U. hordei* hybrids required the expression of effector complements [23]. Therefore, while the action of the constitutively expressed effectors may have been reduced in the *U. maydis* × *S. reilianum* hybrid this was likely not the only issue with infection. There may be a requirement for coordinated expression of multiple effectors to yield more successful hybrid infections.

The expression of transcription factors has the potential to influence the expression of multiple effectors. In *U. maydis, rbf1* acts as a master regulator influencing the expression of 90% of the bE/bW responsive genes [62,63]. When this gene is deleted in *U. maydis* virulence is strongly impaired. In the hybrid, *rbf1* transcript levels were reduced and *srrbf1* transcripts were detected only at 3 and 7 DPI. The expression of *rbf1* indicates that the bE/bW heterodimer is functional suggesting that other *b*-induced genes are also expressed. However, the alterations in transcript levels and timing of *rbf1* expression may have altered expression of downstream genes, or Rbf1 interactions with other *b*-induced genes. The constitutive expression of *rbf1* in the hybrid caused increased filamentous growth on plates, consistent with previous investigations that proposed *rbf1* influences filamentous growth [62]. During infection, hybrids constitutively expressing *rbf1* induced anthocyanin production over larger areas of the leaf surface and resulted in the rare formation of tumors. This indicates that *rbf1* expression led to a gain of function by the hybrid resulting in a localized infection, an attribute of the *U. maydis* dikaryon.

In *U. maydis,* the deletion of *hdp2* does not impact filamentation on plates but is responsible for appressoria formation and driving the expression of an early wave of effectors during infection [24,64]. Our data indicates that constitutive expression of *hdp2* negatively impacted *U. maydis* and hybrid filament formation on plates which suggests a previously unrecognized role for Hdp2 in suppressing genes involved in filament formation. In the hybrid, *hdp2* was only expressed at 1 DPI suggesting that genes downstream of *hdp2* in the *U. maydis* nucleus were not being properly regulated. The hybrid expressing *hdp2* had an increased infection rate and virulence including the limited induction of small leaf tumors. The increased infection rate and virulence was unexpected given the reduced filamentous growth on plates. It could be that the constitutive expression of *hdp2* impacts filamentous growth in a distinct manner depending on whether the dikaryon is growing on a plate or in *Z. mays.* Another possibility is that the constitutive expression of *hdp2* during growth in the plant increased appressorium formation resulting in a higher level of plant cell penetration. Cross sections of infected plant tissue suggested that the amount of fungal colonization was not impacted with *hdp2* expression. However, the level of appressoria formation was not assessed and future analyses could include quantifying appressoria [65]. It is intriguing that the *hdp2* transcript is not detected in the hybrid when expressed from its native promoter and location after 1 DPI or after 3 DPI when constitutively expressed from an ectopic location in the genome. Ectopic expression of the other five *U. maydis* genes using the same otef promoter and inserted into the same region of the genome were expressed. The *hdp2* gene in *U. maydis* has a long natural antisense transcript (NAT) while its ortholog in *S. reilianum* has a short NAT [66]. *U. maydis* does not have RNAi but *S. reilianum* does and it could be that the *hdp2* NAT influences the level of the *hdp2* mRNA in the hybrid by a mechanism that is not present in *U. maydis*. Notably, *tin2* and *nlt1* also have complementary NATs in *U. maydis*, and their transcripts were either detected late or not at all in the hybrid, but the transcripts of these genes were detected when expressed ectopically. Regardless of the reason, the inability to detect *hdp2* mRNA after a certain point would be expected to influence the expression of downstream genes including the early wave of effectors and this could be related to the block in pathogenic development in the hybrid and the limited ability of the hybrid constitutively expressing *hdp2* to induce tumor formation.

The *U. maydis nlt1,* identified as a late transcription factor whose role is to drive expression of a subset of late wave effectors and to induce leaf tumor formation, was not detected in either nucleus of 521 × SRZ2. When 521 constitutively expressing *nlt1* was crossed with either 518 or SRZ2, filament formation on plates was severely reduced indicating it has a role in the regulation of filament formation. Previous work by Lanver et al. [24] showed that disruption of *nlt1* blocked the ability of *U. maydis* to form leaf tumors but infections with these strains induced strong anthocyanin production in the leaves. This was consistent with the 521 × SRZ2 infections in which *nlt1* was not detected and there were no tumors, but anthocyanin induction was observed. Inoculations with the hybrid constitutively expressing *nlt1* resulted in reduced anthocyanin induction but still no tumor formation. There was pronounced chlorosis, which may suggest a hypersensitive response was triggered, and it has been proposed that chlorosis and small necrotic spots in *U. maydis* infections are an outcome of unsuccessful penetration events [28,67,68]. Previous work into *U. maydis* dikaryons constitutively expressing *nlt1* found lower infection rates and reduced virulence (Saville et al., unpublished). The negative impact on pathogenic development by the constitutive expression of *nlt1* in the hybrid and in *U. maydis* may have caused expression of effectors that are usually expressed later in pathogenesis. The plant could have then detected this expression and possibly triggered a hypersensitive response that inhibited further pathogenic development. This may suggest that appropriately timed expression of *nlt1* is required for it to improve pathogenic development and cause tumor formation in the hybrid.

The positive influence of constitutively expressing individual transcription factors in the hybrid during its pathogenic development, opens avenues for future investigation. Finding ways to enhance the virulence of these hybrids constitutively expressing *rbf1* or *hdp2* such as inoculating different maize varieties or using different methods of inoculation could provide an ability to investigate tumors and teliospore formation in the hybrid. Overall, the data presented here indicate that *U. maydis* × *S. reilianum* hybrids can infect *Z. mays* and that manipulation of gene expression in these hybrids can alter their virulence. This work provides a proof of concept that this system can be used to investigate smut fungal hybridization and the emergence of new pathotypes.

## 5. Conclusions

In this study, the successful formation of *U. maydis* × *S. reilianum* hybrids provides the basis for studying hybrid pathogen emergence using the extensive *U. maydis* tool set and the ability to compare the hybrid infection to those of each well studied parental dikaryon. In seedling pathogenesis assays the hybrid was capable of colonizing the plant and inducing anthocyanin production but did not exhibit extensive hyphal growth nor progress in pathogenic development. It was discovered that constitutively expressing transcription factors with roles in *U. maydis* pathogenesis enhanced the virulence of the hybrid, while expressing individual effectors did not. This indicates that this hybrid system can be used to explore pathogen emergence through the expression of virulence genes and assessing possible gains of function. Moving forward, it will be interesting to express more than one virulence gene and potentially determine a minimum number of transcriptional changes that are required for new pathogen emergence. Future investigations using the hybrid will also provide insight into which gene or gene sets influence the localized mode of infection and perhaps uncover alterations that enabled smut fungi to gain the ability to create localized infections. The development of a *U. maydis* × *S. reilianum* hybrid system provides a proof of concept for investigating gene function in a hybrid pathogen and suggests a novel way to investigate smut fungal evolution.

## Figures and Tables

**Figure 1 jof-07-00672-f001:**
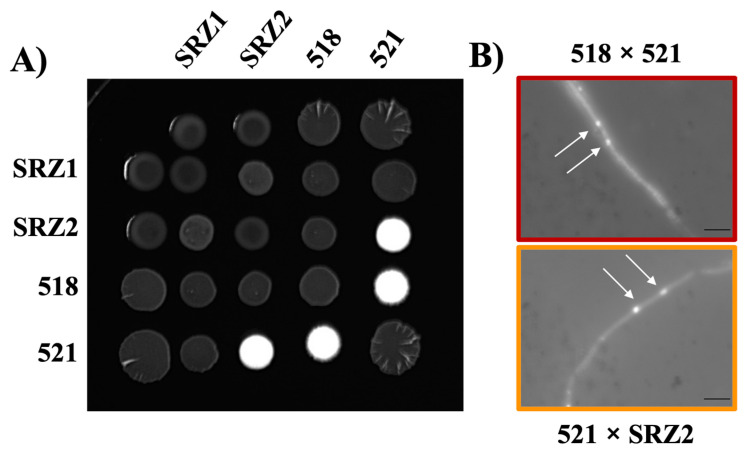
Dikaryon formation by 521 and SRZ2. The indicated mixtures of compatible haploid strains were spotted onto PDA + charcoal plates and incubated for three days. (**A**) Co-spotting of indicated haploid strains. White fuzz indicated the formation of filaments (**B**) DAPI stained filaments of 518 × 521 and 521 × SRZ2 revealed the presence of two nuclei, indicated by the arrows. Scale bar, 2 μm.

**Figure 2 jof-07-00672-f002:**
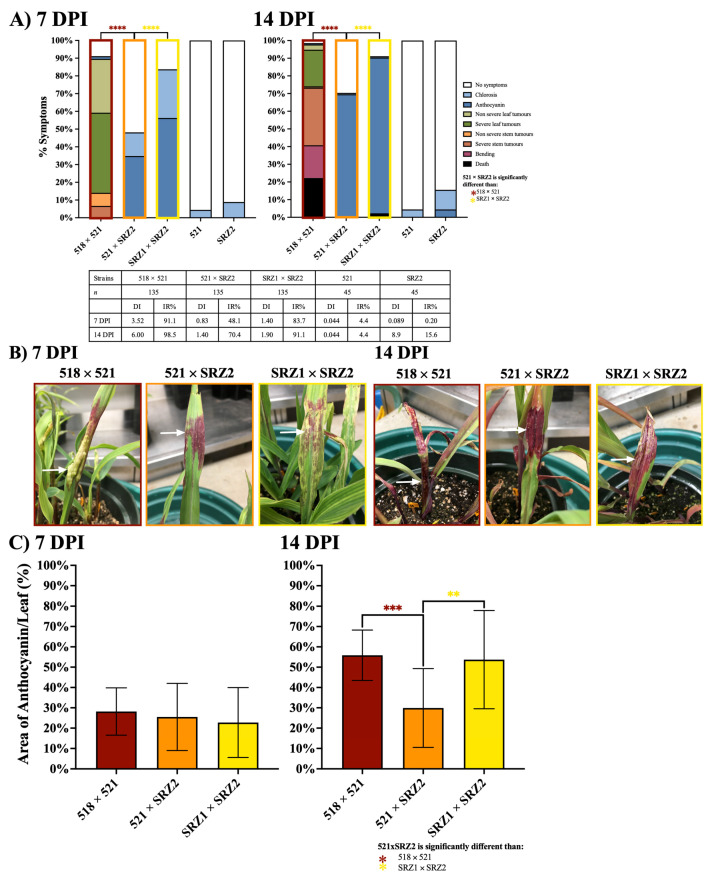
Hybrid infection induced a distinct anthocyanin induction phenotype at 7 and 14 DPI. (**A**) Colored bars represent the percentage of plants that developed a scored symptom, as indicated in the legend. The table presents the number of plants infected (*n*), disease index (DI), and infection rate (IR). (**B**) Representative symptoms, indicated by the arrows, for 518 × 521 infections are leaf tumors at 7 DPI and stem tumors at 14 DPI, for 521 × SRZ2 infections is anthocyanin at 7 and 14 DPI, and for SRZ1 × SRZ2 infections are anthocyanin + chlorosis at 7 and 14 DPI. (**C**) The percent (%) of leaf surface area in which anthocyanin was induced was quantified using ImageJ software. Three separate experiments were conducted, and the combined results are presented. Statistical differences were calculated comparing the hybrid to each parental dikaryon (** = *p* < 0.01; *** = *p* < 0.001; **** = *p* < 0.0001).

**Figure 3 jof-07-00672-f003:**
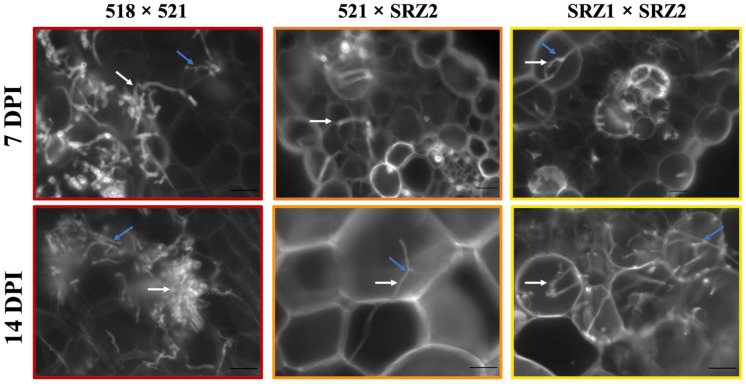
Hyphal growth of 521 × SRZ2 was low compared to 518 × 521 and SRZ1 × SRZ2 at 7 and 14 DPI. Representative cross sections of infected *Z. mays* tissue stained with Fungi-Fluor and viewed under Zeiss microscope at 400×. White arrows indicate hyphal growth through plant cells and blue arrows indicate hyphal branching. Scale bar, 20 μm.

**Figure 4 jof-07-00672-f004:**
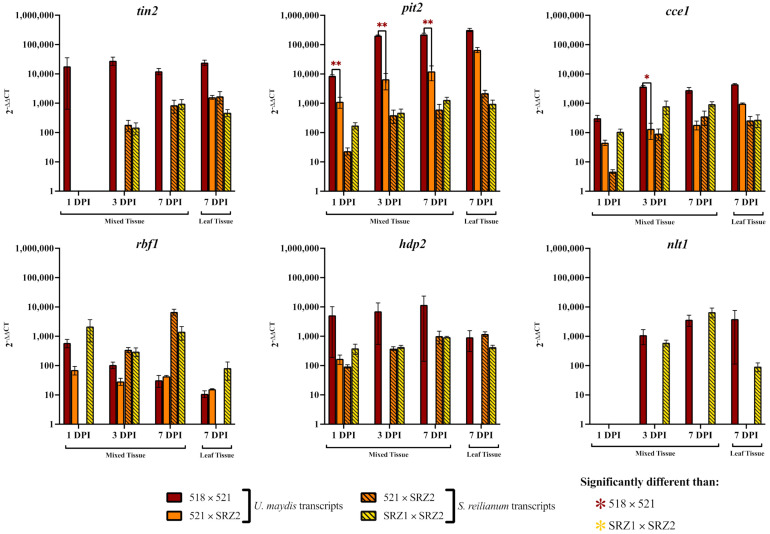
Transcript levels, determined by RT-qPCR, of effectors and transcription factors compared in *U. maydis*, hybrid, and *S. reilianum* dikaryon infections of *Z. mays*. RNA was isolated from infected plant tissue harvested at 1, 3, and 7 DPI. RT-qPCR analysis was performed using three technical replicates for each sample and used the housekeeping gene (*gapdh*) transcript levels for normalization. Relative expression was determined using the 2^−^^ΔΔCt^ method with the axenically grown haploid (521 or SRZ2) for reference. The average and SEM (error bars) of three and two biological replicates is reported for mixed and pure leaf tissues, respectively. Significant difference between 521 × SRZ2 and each parental dikaryon was determined by Student *t*-tests (* = *p* < 0.05; ** = *p* < 0.01).

**Figure 5 jof-07-00672-f005:**
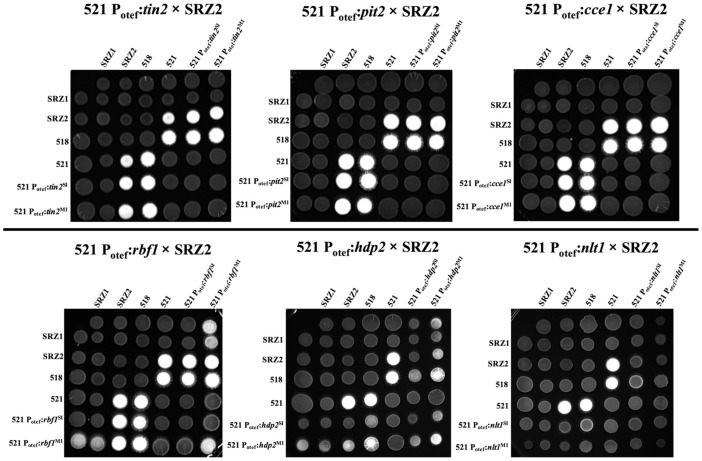
Constitutively expressed transcription factors altered filament formation, while effector expression did not. The indicated mixtures of compatible haploid strains were spotted onto PDA + charcoal plates and incubated for three days in the dark. White fuzz indicates the formation of filaments.

**Figure 6 jof-07-00672-f006:**
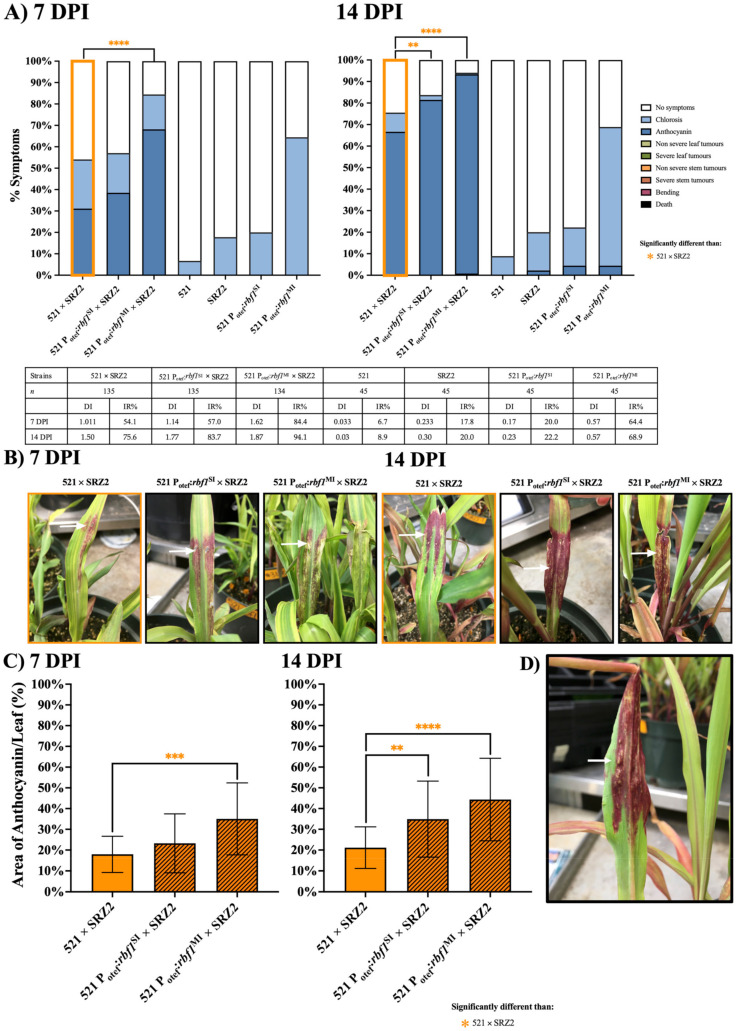
Hybrids constitutively expressing *rbf1* induced greater anthocyanin than the unaltered hybrid at 7 and 14 DPI. (**A**) Colored bars represent the percentage of plants that developed a scored symptom, as indicated in the legend. The table presents the number of plants infected (*n*), disease index (DI), and infection rate (IR). (**B**) Representative symptoms, indicated by the arrows, for 521 × SRZ2 infections is anthocyanin, for 521 P_otef_:*rbf1*^SI^ × SRZ2 infections is anthocyanin, and for 521 P_otef_:*rbf1*^MI^ × SRZ2 infections are anthocyanin + chlorosis. (**C**) The percent (%) of leaf surface area in which anthocyanin was induced was quantified using ImageJ software. Three separate experiments were conducted, and the combined results are presented. Statistical differences were calculated comparing the hybrid to each parental dikaryon (** = *p* < 0.01; *** = *p* < 0.001; **** = *p* < 0.0001). (**D**) Leaf tumor formation, indicated by the arrow, induced by the 521 P_otef_:*rbf1*^MI^ × SRZ2 infection.

**Figure 7 jof-07-00672-f007:**
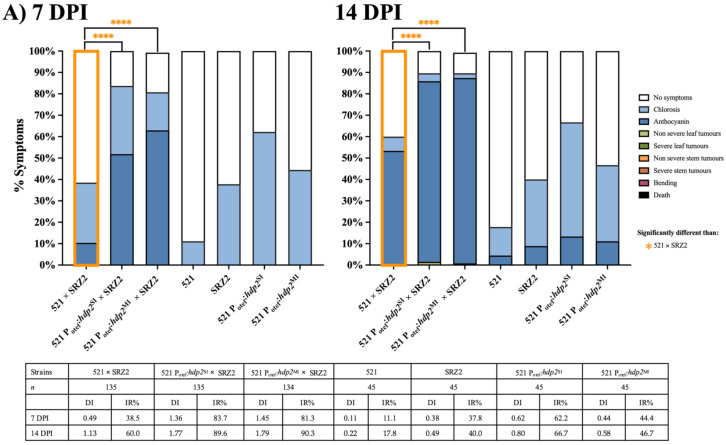
Hybrids constitutively expressing *hdp2* induced greater anthocyanin than the unaltered hybrid at 7 and 14 DPI. (**A**) Colored bars represent the percentage of plants that developed a scored symptom. The table presents the number of plants infected (*n*), disease index (DI), and infection rate (IR). (**B**) Representative symptoms, indicated by the arrows, for 521 × SRZ2 infections is anthocyanin, for 521 P_otef_:*hdp2*^SI^ × SRZ2 infections are anthocyanin + chlorosis, and for 521 P_otef_:*hdp2*^MI^ × SRZ2 infections are anthocyanin + chlorosis. (**C**) The percent (%) of leaf surface area in which anthocyanin was induced was quantified using ImageJ software. Note: only six leaves were processed for 521 × SRZ2 at 7 DPI. Three separate experiments were conducted, and the combined results are presented. Statistical differences were calculated comparing the hybrid to each parental dikaryon (**** = *p* < 0.0001). (**D**) Leaf tumor formation, indicated by the arrow, induced by the 521 P_otef_:*hdp2*^SI^ × SRZ2 infection.

**Figure 8 jof-07-00672-f008:**
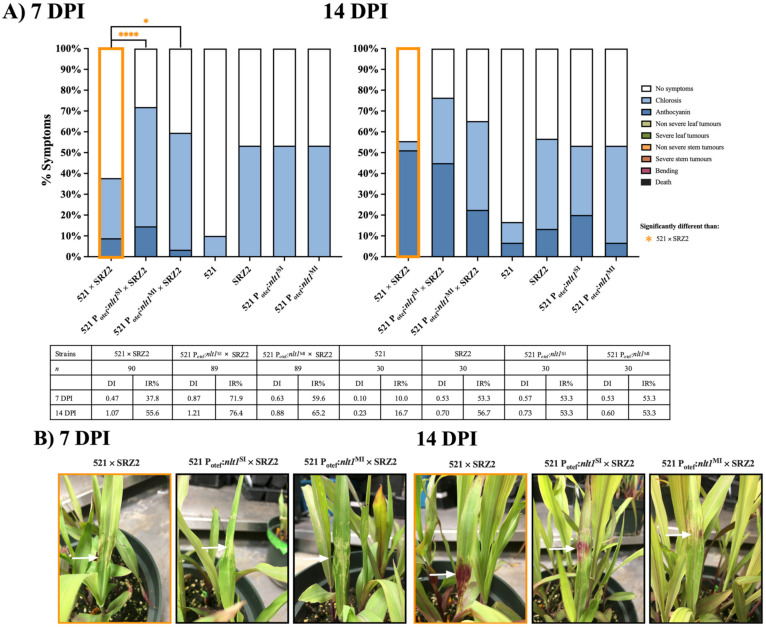
Infections with hybrids constitutively expressing *nlt1* produced pronounced chlorosis at 7 and 14 DPI. (**A**) Colored bars represent the percentage of plants that developed a scored symptom. The table presents the number of plants infected (*n*), disease index (DI), and infection rate (IR). (**B**) Representative symptoms, indicated by the arrows, for 521 × SRZ2 infections are chlorosis at 7 DPI and anthocyanin at 14 DPI, for 521 P_otef_:*nlt1*^SI^ × SRZ2 infections are chlorosis at 7 DPI and anthocyanin at 14 DPI, and for 521 P_otef_:*nlt1*^MI^ × SRZ2 infections are chlorosis at 7 DPI and chlorosis + anthocyanin at 14 DPI. Two separate experiments were conducted, and the combined results are presented. Statistical differences were calculated comparing the hybrid to each parental dikaryon (* = *p* < 0.05; **** = *p* < 0.0001).

**Table 1 jof-07-00672-t001:** Fungal strains used in this study.

Haploid Strain ^a^	Relevant Genotype	Source
518	a2b2	Holliday [32]
521 ^b^	a1b1	Holliday [32]
SRZ1	a1b1	Schirawski et al. [19]
SRZ2 ^b^	a2b2	Schirawski et al. [19]
521 P_otef_:*rbf1*^SI^	a1b1 ip^R^[P_otef_:*rbf1*^SI^]ip^S^	This Study
521 P_otef_:*rbf1*^MI^	a1b1 ip^R^[P_otef_:*rbf1*^MI^]ip^S^	This Study
521 P_otef_:*hdp2*^SI^	a1b1 ip^R^[P_otef_:*hdp2*^SI^]ip^S^	This Study
521 P_otef_:*hdp2*^MI^	a1b1 ip^R^[P_otef_:*hdp2*^MI^]ip^S^	This Study
521 P_otef_:*nlt1*^SI^	a1b1 ip^R^[P_otef_:*nlt1*^SI^]ip^S^	Meade et al. [unpublished] This Study
521 P_otef_:*nlt1*^MI^	a1b1 ip^R^[P_otef_:*nlt1*^MI^]ip^S^	Meade et al. [unpublished] This Study
521 P_otef_:*tin2*^SI^	a1b1 ip^R^[P_otef_:*tin2*^SI^]ip^S^	Cheung et al. [33] This Study
521 P_otef_:*tin2*^MI^	a1b1 ip^R^[P_otef_:*tin2*^MI^]ip^S^	Cheung et al. [33] This Study
521 P_otef_:*pit2*^SI^	a1b1 ip^R^[P_otef_:*pit2*^SI^]ip^S^	This Study
521 P_otef_:*pit2*^MI^	a1b1 ip^R^[P_otef_:*pit2*^MI^]ip^S^	This Study
521 P_otef_:*cce1*^SI^	a1b1 ip^R^[P_otef_:*cce1*^SI^]ip^S^	This Study
521 P_otef_:*cce1*^MI^	a1b1 ip^R^[P_otef_:*cce1*^MI^]ip^S^	This Study

^a^ Single insert (SI) and multiple insert (MI), ^b^ strain used for genome sequencing [26,27].

## Data Availability

Available upon request.

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
