# Peer review of "Fungal Pathogen Emergence: Investigations with an Ustilago maydis × Sporisorium reilianum Hybrid"

_jof, 2021, doi:10.3390/jof7080672_

Round 1

Reviewer 1 Report

I thank the authors for considering my comments to improve the manuscript. The authors respond most of my comments adequately, and I think the manuscript is now worth to be published.

Reviewer 2 Report

Thank you for the detailed and careful changes in the resubmitted manuscript.

It is now a pleasure to read this interesting documentation about fungal pathogen emergence.

This manuscript is a resubmission of an earlier submission. The following is a list of the peer review reports and author responses from that submission.

Round 1

Reviewer 1 Report

Comments for authors

This manuscript attempts to raise a case of an emerging pathogen using the U. maydis X S. reilianum hybrid as a study system. The authors forced to generate the hybrid in axenic cultures and used it as an inoculum for an infection assay. The hybrid can grow and colonize in the plant host (Z. mays), but with less virulence than non-hybrid dikaryons. The authors then investigate transcript level of a few genes involved in virulence, followed by ectopic expression to evaluate if they are involved in virulence and pathogenesis.

Major comments

Overall, I appreciate the significant amount of work in infection assays performed by the authors. However, there are lots of missing/not shown data that should critically support for discussion and conclusion inferred by the authors. I do not think I am convinced with the conclusion based on the data provided in this manuscript. Here are a few examples of what I mean:

- The authors discuss that the hybrid neither has late pathogenic development nor teliospore formation. I did not see any data on this in the manuscript.

- The authors mention about ‘distinct’ anthocyanin phenotype in the hybrid, but not clearly state in detail how. Even the authors mention ‘more/less anthocyanin pigmentation’, the authors do not show any quantification data expect a few photographs, which are hard to make an inference.

- The authors state that the hybrid has fewer hyphal branching then the non-hybrid dikaryons. This could not be judged based on a few photomicrographs. Quantification by counting a significant amount of hyphal cells would be critical to support the statement.

- The authors discuss about transcript levels of the hybrid compared to non-hybrid dikaryons. The authors use the word like ‘higher/lower/comparable. However, no statistical test is shown to support the statement. Why do not the authors use dikaryon grown in axenic cultures as a reference for RT-qPCR quantification? Otherwise, how do the authors assure that the elevation of the six transcripts is due to plant colonization, not due to dikaryotization of the smut fungi?

- The authors spend several paragraphs explaining how the hybrid has a different pathogenic development. However, transcript levels of six selected genes are shown as supporting data. The fact that ectopic expression of these genes result in more severity does not explain the pathogenic development of the hybrid as these genes have been already shown to affect the virulence of U. maydis. I think the discussion part on pathogenic mechanism should be truncated unless the authors have strong supporting data.

- To prove the concept of ‘emergence pathogen’, it would be great to inoculate both haploid U. maydis and haploid S. reilianum on the same host and see if pathogenic development can occur. Is there any research conducted in this area?

- The materials and methods are not easy to follow. For example, the authors’ writing is not clear on when to use liquid cultures or agar cultures.

- There are many places for ‘data not shown’. As there is an unlimited space for supplementary materials, I strongly recommend the authors to put those ‘data not shown’ in the supplement. Some of those are important to understand the rationale of some experiments/analyses.

- The authors are not clear on how they downselect to six genes for examining transcript levels. This would be really important for the readers to understand the rationale of the study. The rationale should be somehow mentioned in the introduction and/or abstract.

- The study of Kellner et al. (2011) illustrates interspecific compatibility through the PR system (MAT a), but not the heterodimer (MAT b). I think it would be great to explain a pathogenic mechanism of the hybrid by starting with a proof to show an active bW/bE heterodimer in the hybrid.

- In the introduction, the authors raise a case of hybrids revealed by genome sequencing. This is not clear to me. For example, the authors mention about UG99 that the pathovar is a hybrid. However, the authors do not explain how it becomes a hybrid, or what the composition of the hybrid is, or how they know that this is a hybrid based on sequencing data.

- Please double-check spelling and grammars throughout the manuscript. I catch some of those while reading. See in minor comments.

Minor comments

- Abstract: Can the authors name some of interesting effectors, transcription factors involved in this study?

- Line 14: Should be ‘Selected’

- Line 30 – 32: Should clarify if the authors are talking about interspecific hybridization.

- Line 39: I recommend putting parentheses after Ustilaginaceae to indicate that this lineage belongs to Basidiomycota. Like “… belonging to Ustilaginaceae (Basidiomycota), …”

- Line 40: Do they need to always target on sweet grasses? What do the authors mean by ‘sweet’?

- Line 43: I would recommend saying ‘mating type loci’, as these loci do not need to have conserved synteny of several genes in a single locus, to be enough to call “gene cluster”. I recommend replacing “mating type clusters” to “mating type loci”. This should be corrected throughout the manuscript.

- Line 51: I would recommend saying “… in which the two mating type loci are located on the same chromosome.

- Line 65 – 66: Can U. maydis induce tumor in all aerial parts? Please clarify.

- Line 69 – 75: I think this can be trimmed down. The authors can just say the objective of the study and the rationale of the study system.

- Line 71: Please check the spelling ‘virulence’.

- Table 1: Check for the consistency of citation style. For example, the authors say Holliday [25], Schirawski et al. [17] and Cheung et al. (2021). I think you can just use only square brackets.

- Materials and methods: indicate city and country of the company headquarter for the first time you mention the company.

- Line 82: I recommended saying ‘In liquid culture, ….’.

-  Line 83 – 84: Why growing on Carboxin? Is it a selectable marker used in the insertion strains? Please clarify here. Also, are expression strains grown on agar plates, in liquid media or both?

- Line 105ff: For how long? 16 and 18 hours as abovementioned for both steps (before and after culture transfer)?

- Line 118: Specify company name for the GraphPad software.

- Line 123: What samples? Fungal samples on plates? Also check grammar ‘a VWR VistaVision glass microscope slides

- Line 126ff: Is 24-hrs DAPI staining too long? Would the sample get dried off? How do the authors to keep it moist?

- Line 138 and 145: Please italicize species name.

- Line 150: Please put references for PROSITE, PFAM and SignalP.

- Section 2.5: Representation as a table may look better? I would recommend the section title as ‘Orthology assessment and sequence analyses’ It would be good to clarify here that you use ‘sr’ as a prefix of gene name to indicate gene ortholog in S. reilianum when performing gene expression analyses.

- Line 154 – 155: For how long? 16 and 18 hours as abovementioned for both steps (before and after culture transfer)? This is a same comment with line 105ff.

- Line 153: I recommend saying “Symptomatic plant tissues” to clarify.

- Line 156ff: Please indicate which reference is for which type of samples (ground frozen plant tissues and fresh fungal pellets).

Line 175: Use the word ‘Construction’ would be a better choice.

- Table A1: c Lower-case letters in primer sequences represent tags containing restriction enzyme sites. Of which enzymes? Please specify.

- Figure 1: Having arrowheads labeling septa would be useful for the readers to judge dikaryon. Also, having focal planes with two or more cell would be great support the statement.

- Figure A2: Personally I think Graphs in Figure A2 can be put as another panel in Figure 1, as it demonstrates that the successful hybrid mating leads to virulence on plant hosts.

- Line 246: Please check the spelling ‘Focusing’

- Line 256: It would be great if the authors have the data for 5 and 7 DPI for infection assays of different seedling ages. Adding them as a panel in Figure A3 would benefit readers to support the statement.

- Line 278: When the authors say ‘a larger surface area’, do the authors use a visual inspection, or measure the area of infestation?

- Figures 1 – 2: I think Figure 1 and 2 be combined as Figure 1 with four panels, as the objective is to show that the hybrid has a stable filamentous growth, and the hybrid can infect the host with the sign of infection.

- Figure A4: Please specify tumor growth in the figure.

- Figures A5, 3: In think Figure A5 and 3 can be combined into a single figure. It would be good to use same magnification for all panels so it would be easier for readers to compared. For example, I do not believe that the scales for 7DPI and 14DPI photomicrographs for 521xSRZ2 are both 20 microns. Some panels like 518x521 at 7DPI and 14DPI do not have a good resolution. Please consider improvement/replacement.

- Line 313ff: I think this piece of data is important as it indicates why the authors downselect to six genes in subsequent experiments. It should be at least in the supplementary material, instead of just saying ‘data not shown’. Also, please clarify what the authors see based on the screens. For example, are transcripts of these six genes are overexpressed or underexpressed in the hybrid compared to haploid strains or non-hybrid dikaryons?

- Line 332: Please specify that ‘when compared to the transcript level of axenic haploid strains’.

- Line 336-337: “but at lower levels and following a pattern that differed from that of 518x521.” How? Please specify.

- Line 340 – 341: How is the pattern different? Please specify. For RT-qPCR, does Delta delta ct value come from a comparison of three transcript replicates? Is there any variation of transcripts in haploid strains?

- Figure 4: Representation in a Y axis as a log2 fold change would be nicer? Is there any statistical test to show ‘higher/lower/compararble’ transcript? If so, the authors should present that in the figure.

- Line 404 – 405: Again, when the authors say “the surface area of anthocyanin on the leaf was consistently greater in both hybrids expressing rbf1 relative to 521xSRZ2”, do the authors perform any quantification to see if there is a statistical significance? Or just do the visual inspection? If the latter, can the authors give the number of observations?

- Figure 6: What is the symptom indicated at an arrowhead? Chlorosis? Please specify.

- Figure 7: I think Figure 7 can be incorporated as another panel in Figure 6. What does the arrowhead indicate? Please specify.

- Line 412: Is there any quantification to illustrate similar amount of hyphal growth? Amount of hyphal growth may depend on the focal/section plane that is used to observe fungi? Do the authors try fluorescence assay, or any way to quantification (for example, look through slides for at least five focal planes and see the same thing, something like this)?

- Figure 9: I think Figure 9 can be incorporated as another panel in Figure 8. Please indicate what an arrowhead means.

- Legends for Figures 2, 6, 8, 10: What do the authors mean by “Two/Three combined biological replicates are presented.” Does it mean that the authors repeat the experiment for 2 – 3 times for statistical analyses? I do not think 2 – 3 biological replicates mean the 2 – 3 plants. Please clarify.

- The tables for Figures 2, 6, 8, 10: A number of infected plants are integer; they should not have decimal points.

- Discussion part: Putting references for figures in each part of the discussion would help readers follow along well.

- Line 483: I would recommend changing to “This suggests that” instead of “Suggesting”. Also, don’t the authors’ result indicate that the virulence is less than the parents? To me, using the words ‘distinct from the parents’ mean that the plant hosts should have remarkably different symptoms. However, the hosts infested by the hybrid still have similar symptoms, but less severity.

- Line 487: I am not sure if the data the authors present (Figure 3, A5) can indicate “limited branching in the host cells”. Better photomicrographs showing the difference in hyphal branching between the hybrid and the parental dikaryons, or any quantification, should be presented. Otherwise, I think the statement that ‘less hyphal branching’ is not convincing. Less proliferation may not be always coupled with less hyphal branching.

- Line 495 – 497: Do the authors mean this current study? If so, I do not see any data showing that the plant hosts infested by S. reilianum can form inflorescences.

- Line 498ff: Again, there is no clear indication about how different in terms of color and coverage among three types of infections.

- Line 506 – 507: I am not sure if the authors can make this claim unless showing data on pigment quantification and coverage of anthocyanin production.

- Line 510 – 512: Data about measuring tumor formation and teliospore formation in the seedlings/inflorescences are not shown. I am not convinced on this statement unless the authors show the data on these aspects.

- Line 515 -516: “because signals produced by the plant were detected or transmitted differently.” I am not sure about possibility of this explanation. Even though there are two different nuclei, they belong to a same cell. Thus, I think the signal detection would be similar, but maybe the difference is at the level of transcriptional machineries for two nuclei.

- Line 516ff: I am not sure if the authors have a strong evidence for this statement. For instance, the authors say “The downregulation and lack of detected U. maydis specific transcripts” (according to Figure 4?), but what about the expression of cce1 which the Sr transcripts in the hybrid is less than the non-hybrid? Also, the missing of Sr rbf1 transcript is also found in the hybrid. Inference based on six genes that do not have consistent patterns would be challenging to make a strong conclusion.

- Line 524ff: I do not see any data in the manuscript that supports this paragraph.

- Line 534 – 536: I am not sure if this statement convinces me as the authors use transcripts level of haploid strains in axenic cultures, not in inoculated tissues, for comparison.

- Line 556ff: When looking through Figure 2, it looks like the hybrid causes anthocyanin production more than 518x521. As the authors say that tin2 is proposed to be responsible for this induction of anthocyanin, how can it explain the fact the Um tin2 transcript is not found in the hybrid? The logic is confusing here. Checking lignin content of plant host would be another piece of evidence to make this statement solid.

- Line 606 – 609: The authors could have checked if bE/bW genes are active in the hybrid. This would be another key evidence to say that the hybrid is pathogenic to plant hosts.

Reviewer 2 Report

In their manuscript entitled „Fungal pathogen emergence: investigations with an Ustilago maydis x Sporisorium reilianum hybrid the authors Storfie and Saville analysed the pathogenic development on planta and collected the expression pattern of different effector and transcription factor genes at specific time points after infection. The work starts with the fusions of compatible wildtype and hybrid combinations. Furtheron the hybrid Um521 and SRZ2 were investigated, which was able to build aerial hyphae on charcoal plates. This hybrid was dramatically reduced in virulence and showed only anthocyanin reaction on the plant. Effector gene expression was reduced in the hybrid when compared to the wildtype but overexpression of one single effector gene did not enhance pathogenic development. The overexpression of b-dependent transcription factors rbf1 and hdp1 led to development of small tumors.

The work is well executed and many data are shown in the manuscript and the supplementals. Therefore, the explanations and descriptions are detailed but finally for me a bit boring. Here a more focused and combined way in analyzing the data would be better.

Finally, they end with the small tumor development. But the authors did not go further. Is there spore development? If there are spores, are they able to germinate? This would be answer the question of real emergence of novel fungal pathogens. At least this should be discussed. In addition, there is a third a-locus in S. reilianum. Have the authors checked the missing mating combinations? If not, Please discuss.

There are some minor points:

Abstract: Selected virulence…

Lane 39: Within fungi is a group…

In general, I was confused by numbering the supplementals with A1)—A14). I would propose to change to S1)- S14).

Lane 240: Please describe the wildtype combinations first, then 521xSRZ2.

Lane 256: please show the data of (data not shown)

Figure 2 and others: the number of infected plants should be 135, not 135,00; please change.

Lane 298/Figure 3: Is the fungal development extensive in SRZ1xSRZ2? In my eyes is less extensive than UM.

Lane 320: What is the context of this sentence: The amino acid sequence….. Can be removed.

Lane 330: …were harder… better: … were more difficult…

Lane 332 and all others: Genes of U. maydis can be named umtin2 as srtin2. This would make the identification easier.

Lane 483: Incomplete sentence: Suggesting the hybrid’s….

Sometimes you use tumour and sometimes tumor. Please check.

There is a lot of “Suggestion” in the text.

Lane 636: Is there appressorium formation inside the plant due to the way of infection? Please discuss.